# Osteocytes: Their Lacunocanalicular Structure and Mechanoresponses

**DOI:** 10.3390/ijms23084373

**Published:** 2022-04-15

**Authors:** Takeshi Moriishi, Toshihisa Komori

**Affiliations:** 1Department of Cell Biology, Nagasaki University Graduate School of Biomedical Sciences, Nagasaki 852-8588, Japan; moriishi@nagasaki-u.ac.jp; 2Department of Molecular Bone Biology, Nagasaki University Graduate School of Biomedical Sciences, Nagasaki 852-8588, Japan

**Keywords:** lacunocanalicular network, osteocyte, mechanical stress, compression, tension, Bcl2, Sp7, osteocyte apoptosis, Sost, Rankl

## Abstract

Osteocytes connect with neighboring osteocytes and osteoblasts through their processes and form an osteocyte network. Shear stress on osteocytes, which is induced by fluid flow in the lacunae and canaliculi, has been proposed as an important mechanism for mechanoresponses. The lacunocanalicular structure is differentially developed in the compression and tension sides of femoral cortical bone and the compression side is more organized and has denser and thinner canaliculi. Mice with an impaired lacunocanalicular structure may be useful for evaluation of the relationship between lacunocanalicular structure and mechanoresponses, although their bone component cells are not normal. We show three examples of mice with an impaired lacunocanalicular structure. Ablation of osteocytes by diphtheria toxin caused massive osteocyte apoptosis, necrosis or secondary necrosis that occurred after apoptosis. Osteoblast-specific Bcl2 transgenic mice were found to have a reduced number of osteocyte processes and canaliculi, which caused massive osteocyte apoptosis and a completely interrupted lacunocanalicular network. Osteoblast-specific Sp7 transgenic mice were also revealed to have a reduced number of osteocyte processes and canaliculi, as well as an impaired, but functionally connected, lacunocanalicular network. Here, we show the phenotypes of these mice in physiological and unloaded conditions and deduce the relationship between lacunocanalicular structure and mechanoresponses.

## 1. Introduction

Osteocytes form a dense network in bone. This network is composed of an intracellular network through osteocyte processes and gap junctions, and an extracellular network through lacunae and canaliculi. The intracellular network connects osteocytes not only to their neighboring osteocytes, but also to osteoblasts on the bone surface. The extracellular network extends to the bone marrow, periosteum and blood vessels in the bone. Osteocytes acquire oxygen and nutrients through the two networks [1]. Osteocyte networks are generally considered as a mechanosensory and a mechanotransduction system and shear stress on osteocytes, which is induced by fluid flow, is an important mechanism for mechanoresponses [2,3,4,5,6,7,8,9].

In lamellar bone, spindle-shaped lacunae are aligned to the direction of collagen fibers and canaliculi are aligned perpendicularly, whereas lacunae in disordered collagen fibers are rounded and the canaliculi are oriented isotropically [10,11,12]. Animal studies, which examined the relationship of lacunocanalicular structure and mechanoresponses, are very limited. In this review, we discuss the differential development of the lacunocanalicular structure in the compression and tension sides of femoral cortical bone in wild-type mice, and the relationship between lacunocanalicular structure and mechanoresponses by comparing three mouse models.

## 2. Differences in the Lacunocanalicular Structure between the Compression and Tension Sides

In the diaphysis of mouse femurs, collagen fibers ran longitudinally, lacunae with spindle shapes were uniformly aligned in the longitudinal direction and canaliculi were aligned in the perpendicular direction to lacunae in the anterior cortical bone (compression side); however, the direction of collagen fibers was disturbed in some regions, especially in the periosteal side of the posterior cortical bone (tension side), and the spherical-shaped lacunae and canaliculi were irregularly aligned in this region [13] (Figure 1). In parallel, the length of lacunae in the longitudinal direction in the periosteal side of the compression side was longer than that in the tension side in males [13] (Figure 1B). A lack of highly organized longitudinal collagen is also observed in the tension side of equine cortical bone in the third metacarpals [14]. However, the difference is not apparent in the femoral diaphysis in humans [15], probably due to the relatively complex loading pattern. Thus, mice are a good animal model for analyzing differences in lacunocanalicular structure between the compression and tension sides (Figure 1A). In both male and female mice, canaliculi in the tension side were thicker than those in the compression side of the femurs. The number of canaliculi in the periosteal side of the compression side was higher than that of the tension side in the femoral cortical bone [13] (Figure 1B). Therefore, the compression side of the cortical bone has more organized collagen fibers, lacunae and canaliculi, and denser and thinner canaliculi than the tension side, especially in male mice. Further, the fluid flow velocity was higher and bone formation was greater in the compression side than that in the tension side in an in vivo loading experiment using mouse tibiae [16].

Why is the lacunocanalicular structure different between the compression and tension sides? The lacunar and canalicular areas are enlarged in lactation and they return to normal after lactation, indicating that osteocytes have the capacity for bone lysis and bone formation [17,18]. There are significant differences in the canalicular thickness, but not in the lacuna area, between the compression and tension sides [13]. As the compression side receives more mechanical stress than the tension side, the strength of mechanical stress may be associated with the capacity of osteocytes for mineral apposition in the canaliculi. However, this does not explain the lack of difference in the lacunar area. An unloading experiment also did not affect the lacuna volume in tibiae [17]. These findings suggest that the difference in the canalicular thickness between the compression and tension sides is not due to the direct effect of mechanical stress. Canaliculi are formed in the processes during the transition of osteoblasts into osteocytes. Osteoblasts in the periosteum and endosteum of the cortical bone in the compression side more actively produce bone matrix than those in the tension side [16]. Thus, it is possible that the transitional osteoblasts into osteocytes in the compression side produce more matrix proteins than those in the tension sides from the processes, leading to the reduced canalicular thickness in the compression side.

The increased canalicular number in the compression side indicates that the connection between the processes of osteoblasts and osteocytes during the transition is greater in the compression side than in the tension side. The newly embedded osteocytes with thinner canaliculi in the compression side will require more connection with osteoblasts to acquire enough nutrition and oxygen for survival. Further, osteoblasts in the compression side may have more processes than those in the tension side to release the produced bone matrix, because the former produces more bone matrix than the latter. These will lead to increases in the connection of the processes of osteoblasts and osteocytes and in the number of canaliculi.

## 3. Osteocyte Death and Bone Resorption

As the lacunocanalicular network is required for osteocyte survival, disturbance of the lacunocanalicular network causes osteocyte apoptosis [13,19]. If lacunae with apoptotic osteocytes are not quickly removed by bone resorption, secondary necrosis of the apoptotic osteocytes will occur, because apoptotic osteocytes embedded in the bone matrix cannot be removed by phagocytes. Necrosis leads to rupture of the cytoplasmic membrane and most of the intracellular content is released into the lacunae and canaliculi. However, it is difficult to discriminate whether osteocytes died by apoptosis or necrosis, because the fragmented DNA, which is detected by TUNEL, remains in the lacunae after secondary necrosis until the lacuna area is removed by remodeling [8,20]. The death of osteocytes induces bone resorption [21,22,23,24]. In the necrosis of osteocytes, immunostimulatory molecules, including danger-associated molecular patterns (DAMPs) such as the HMGB1 protein, S100 family molecules, heat-shock proteins, purine metabolites and uric acid, are released from lacunae to the bone surface and the canals of blood vessels through canaliculi (Figure 2). Released DAMPs promote the production of proinflammatory cytokines, including tumor necrosis factor-α (TNF-α), interleukin (IL)-6 and IL-1, which induce Rankl expression in osteoblasts and lead to osteoclastogenesis [8,25,26,27,28,29]. Apoptotic osteocytes also induce osteoclastogenesis. Apoptotic osteocytes release ATP through Panx1 channels [30]. The released ATP recruits macrophages and monocytes, increases Rankl expression in the neighboring osteocytes and osteoblasts, enhances the membrane fusion of osteoclast precursor cells to form multinucleated osteoclasts and increases osteoclast survival [24,29,31,32,33,34] (Figure 2).

## 4. Animal Models for the Investigation of the Relationship between Lacunocanalicular Structure and Mechanoresponses

### 4.1. Ablation of Osteocytes by Diphtheria Toxin

There is no ideal animal model for studying the relationship between lacunocanalicular structure and mechanoresponses, because it is impossible to obtain animals that have abnormal lacunocanalicular structure but a normal number of osteoblasts, osteoclasts and osteocytes, all of which are functionally normal. The first example is the ablation of osteocytes by diphtheria toxin in transgenic (tg) mice expressing the diphtheria toxin receptor under the control of the DMP1 promoter, which directs the expression to osteocytes and terminally differentiated osteoblasts [35] (Figure 2). Diphtheria toxin induced apoptosis or necrosis in osteocytes and terminally differentiated osteoblasts. As most osteocytes died by apoptosis, necrosis or secondary necrosis, bone resorption was markedly enhanced and the necrotic bone was replaced by bone remodeling as previously reported [21,22,23,24]. Further, TNF-α and IL-1, which are induced by DAMPs, activate the NF-κB signaling pathway and inhibit osteoblast differentiation and bone formation [36,37,38,39]. Thus, osteocyte ablation by diphtheria toxin enhanced bone resorption and inhibited bone formation [29,35] (Figure 2 and Figure 4).

An unloading experiment was performed using osteocyte-ablated mice and the enhanced bone resorption and reduced bone formation induced by osteocyte death were not further enhanced by unloading, suggesting that osteocytes are required for the response to unloading [35]. However, the unresponsiveness to unloading is likely due to enhanced bone resorption and reduced bone formation, which were both induced by osteocyte death (Figure 4). Further, unloading followed by reloading reversed the enhanced bone resorption and reduced bone formation in osteocyte-ablated mice, suggesting that osteocytes are not necessary for recovery [35]. However, this is likely the recovery process from the enhanced bone resorption and reduced bone formation by osteocyte death. Although massive osteocyte necrosis completely disrupts the osteocyte network and impairs lacunocanalicular structure, it is not an appropriate model for the evaluation of osteocyte functions and the relationship between lacunocanalicular structure and mechanoresponses, because the bone resorption and bone formation were severely affected by the osteocyte death [40] (Figure 2 and Figure 4). Thus, the osteocyte-ablation mouse is an appropriate model for examining the effects of osteocyte death.

### 4.2. Bcl2 tg Mice

The second example is osteoblast-specific Bcl2 tg mice under the control of the 2.3 kb Col1a1 promoter, which directs the transgene expression to osteoblasts [19,20] (Figure 3C). In Bcl2 tg mice, 80% of the lacunae were positive for TUNEL by 4 months of age. Apoptosis is due to a reduction in the number of osteocyte processes and canaliculi, which is likely caused by a complex formation of Bcl2, actin and gelsolin that impairs the gelsolin-severing activity to increase actin polymerization [19,20,41]. As osteocyte processes and canaliculi are necessary for acquiring oxygen and nutrition, their reduction causes osteocyte apoptosis, although the frequency of osteocyte apoptosis depends on the extent of the reduction. As TUNEL-positive lacunae gradually increases in Bcl2 tg mice, reaching up to 80%, most TUNEL-positive osteocytes should undergo secondary necrosis [20]. However, bone resorption was reduced and bone remodeling did not occur in Bcl2 tg mice. As co-culture of primary osteoblasts from Bcl2 tg mice or Bcl2-overexpressed wild-type primary osteoblasts with bone marrow-derived monocyte/macrophage lineage cells showed normal osteoclastogenesis, reduced bone resorption was likely due to disruption of the lacunocanalicular network [19,20]. This disrupted lacunocanalicular network interrupts the release of DAMPs and ATP from necrotic osteocytes and apoptotic osteocytes, respectively, to the bone surface and vascular channels, and impairs the removal of dead osteocytes by bone resorption. It enabled us to observe the functions of the osteocyte network without the effects of osteocyte death in Bcl2 tg mice (Figure 3C and Figure 4). The reduction in bone resorption in Bcl2 tg mice suggests that the lacunocanalicular network positively regulates bone resorption under physiological conditions.

Primary osteoblasts from Bcl2 tg mice showed normal osteoblastogenesis in vitro, but bone formation was increased in Bcl2 tg mice at 4 months of age [20]. Although the numbers of Sost-positive osteocytes were similar between wild-type and Bcl2 tg mice, the Sost protein could not reach the bone surface due to the disrupted lacunocanalicular structure (Figure 3C). This is one of the reasons for the increased bone formation in Bcl2 tg mice [20].

In an unloaded condition by tail suspension, the bone volume of femurs was reduced in both wild-type and Bcl2 tg mice at 6 weeks of age, at which time the frequency of TUNEL-positive lacunae was 20% in Bcl2 tg mice. Whereas, at 4 months of age, the bone volume of femurs was reduced by unloading in wild-type mice, but not in Bcl2 tg mice, at which time the frequency of TUNEL-positive lacunae was 80% in Bcl2 tg mice. In unloading at 4 months of age, bone resorption was increased in wild-type mice, but not in Bcl2 tg mice, and Rankl expression in osteoblasts was increased in wild-type mice but not in Bcl2 tg mice (Figure 3A–D). This suggests that the osteocyte network is involved in the positive regulation of Rankl expression in osteoblasts in an unloaded condition. Sost expression in osteocytes is downregulated by loading and upregulated by unloading [42,43,44]. Further, Sost-deficient mice are resistant to unloading-induced bone loss and do not show reduced bone formation following unloading [43]. Moreover, the reduction in Sost expression in osteocytes is required for enhanced bone formation by mechanical loading [45]. Sost-positive cells were increased by unloading in the compression side, but not in the tension side, of tibiae in wild-type mice, while they were unchanged in both sides in Bcl2 tg mice, suggesting that the lacunocanalicular network is required for Sost regulation [20] (Figure 4). It is likely that Sost expression is suppressed in a loaded condition more strongly in the compression side than the tension side, and unloading releases the suppression of Sost expression in the compression side. These findings suggest that the lacunocanalicular network responds to unloading, at least in part, through the upregulation of Rankl expression in osteoblasts and Sost expression in osteocytes. As bone resorption was reduced and bone formation was increased in Bcl2 tg mice under physiological conditions, the lacunocanalicular network likely negatively regulates bone mass under physiological conditions, and enhances negative regulation in unloaded conditions by increasing Rankl expression in osteoblasts and Sost expression in osteocytes [8,20]. However, it remains to be clarified how the osteocyte network regulates Rankl expression in osteoblasts. The Bcl2 tg mouse line is not an ideal model for examining osteocyte functions and the relationship between lacunocanalicular structure and mechanoresponses, because the 2.3 kb Col1a1 promoter activity changes during aging, affecting osteoblast proliferation and differentiation and the frequency of dead osteocytes [40]. As the effects of Bcl2 transgene expression on osteoblast proliferation and differentiation were minimal, but the frequency of dead osteocytes was maximal, at 4 months of age [20], Bcl2 tg mice at 4 months of age could be an appropriate model for the analysis of osteocyte functions and the relationship between lacunocanalicular structure and mechanoresponses.

### 4.3. Sp7 tg Mice

The third example is osteoblast-specific Sp7 tg mice under the control of the 2.3 kb Col1a1 promoter [13] (Figure 3E). The number of canaliculi in Sp7 tg mice was about half that of wild-type mice and 14% of lacunae were TUNEL-positive at 14 weeks of age. In contrast to Bcl2 tg mice at 4 months of age, bone resorption was increased and cortical bone was thin and porous. However, trabecular bone volume and bone formation in trabecular and cortical bone were normal, and osteoblast marker gene expression was increased in Sp7 tg mice. The pores in cortical bone contained many osteoclasts and porous cortical bone is likely to be caused by enhanced bone resorption. Osteocyte death, which is likely to be caused by a reduction in the number of canaliculi, is one of the causes of enhanced bone resorption. Collagen fibers and lacunae were irregularly oriented and lacunae were rounded in shape. Canaliculi in Sp7 tg mice were thicker than wild-type mice, and those in females were thicker than those in males. Although the serum level of Sost was reduced in Sp7 tg mice, Sost was disseminated through the canaliculi and reached the bone surface. Thus, the lacunocanalicular network in Sp7 tg mice was functionally connected in the cortical bone [13].

In an unloaded condition by tail suspension, bone loss occurred in both trabecular and cortical bone at a similar or even more severe level in Sp7 tg mice, compared with wild-type mice, due to reduced bone formation [13] (Figure 3A,B,E,F). The serum level of Sost was increased by unloading in Sp7 tg, but not in wild-type mice. Consistent with a previous report [20], Sost-positive cells were increased in the compression side, but not in the tension side, of the femoral cortical bone in wild-type mice, whereas they were marginally increased in both sides of Sp7 tg mice [13] (Figure 4). During unloading for 2 weeks, osteoblast marker gene expression was reduced during the first week, but recovered to the level of the ground group at the end of unloading in wild-type mice. Whereas, osteoblast marker gene expression was markedly decreased at the end of unloading in Sp7 tg mice. Therefore, the lacunocanalicular network, which was markedly impaired but functionally connected, in Sp7 tg mice was sufficient to respond to unloading, at least in part, through the induction of Sost [13] (Figure 3E,F). However, the differential response in the compression and tension sides was lost and the capacity to recover reduced osteoblast marker gene expression during unloading was absent (Figure 4). These findings suggest that an organized lacunocanalicular network is required for the differential regulation of Sost in the compression and tension sides, and the recovery of osteoblast marker gene expression during unloading [13] (Figure 4). Sp7 tg mice are also not an ideal model, but are a useful model, for examining the relationship between lacunocanalicular structure and mechanoresponses. A disadvantage of this mouse is the enhanced bone resorption, probably due to the increase in osteocyte death, although the 14% osteocyte death did not appear to affect bone formation [13] (Figure 3A,E and Figure 4). Indeed, Sp7 overexpression in osteoblasts is likely to have some effect on osteoblasts, osteocytes and osteoclasts, and this should be considered in the evaluation of the relationship between lacunocanalicular structure and mechanoresponses.

## 5. Conclusions

The differences in lacunocanalicular structure between compression and tension sides suggest that mechanical stress affects the development of the lacunocanalicular structure. The relationship between the lacunocanalicular structure and mechanoresponses can only be examined in vivo; however, there are no ideal animal models because bone component cells are altered in animals with an abnormal lacunocanalicular structure. Therefore, the only way to deduce this relationship is to observe the whole phenotypes, under physiological and loaded/unloaded conditions, in animals with abnormal lacunocanalicular structure, although it should be understood that the altered number and/or functions of bone component cells affect mechanoresponses. Further accumulation of data from such animals will help to clarify the relationship between the structure and functions of the lacunocanalicular network and mechanoresponses.

## Figures and Tables

**Figure 1 ijms-23-04373-f001:**
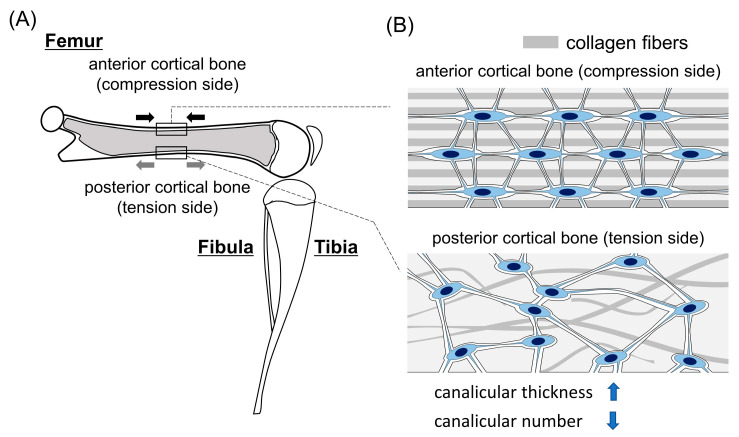
Differences in lacunocanalicular structure between the compression and tension sides of femoral cortical bone: (**A**) A schematic lateral view of the hindlimb of mice. The femur is shown as a sagittal section. Under physiological posture and movement, the anterior and posterior cortical bones receive compression (black arrows) and tension (gray arrows), respectively, in the femur. (**B**) The lacunocanalicular structure of anterior and posterior cortical bones at diaphysis of the femur. Anterior cortical bone has more organized collagen fibers and lacunae, and thinner and more canaliculi than posterior cortical bone.

**Figure 2 ijms-23-04373-f002:**
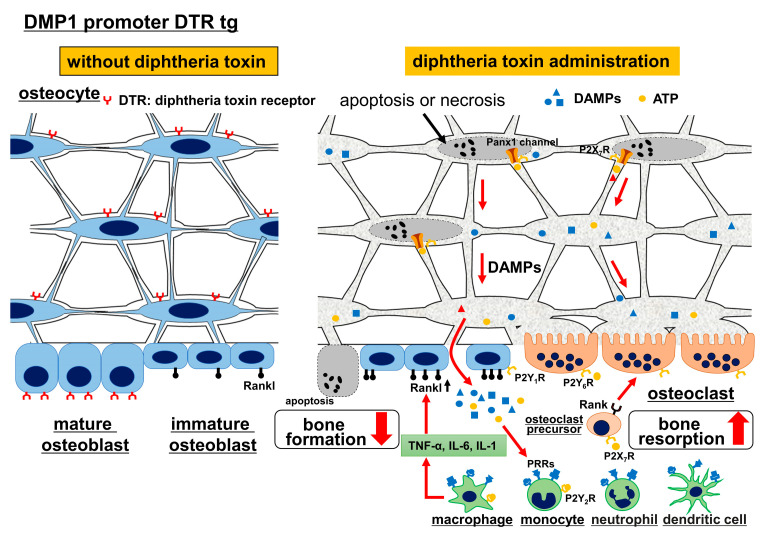
Ablation of osteocytes by diphtheria toxin. In diphtheria toxin receptor (DTR) tg mice under the control of the DMP1 promoter, administration of diphtheria toxin causes apoptosis, necrosis or secondary necrosis in osteocytes and terminally differentiated osteoblasts. DAMPs (danger-associated molecular patterns) are released from necrotic cells and stimulate macrophage, monocytes, dendritic cells and neutrophils through PRRs (pattern recognition receptors), leading to the production of TNFα, IL-6 and IL-1, which induce bone resorption by stimulating Rankl expression in osteoblasts. Further, activation of NF-kB signaling by inflammatory cytokines inhibits bone formation. Apoptotic osteocytes release ATP. Extracellular ATP binds to P2YRs (P2Y G-protein-coupled receptors) and P2XRs (P2X ligand-gated ion channels), recruits macrophages and monocytes through P2Y2R, enhances Rankl expression in osteocytes through P2X7R and in osteoblasts through P2Y1R, increases osteoclast survival through P2Y6R, induces membrane fusion of osteoclast precursors through P2X7R, enhances ATP release from Panx1 channels in osteocytes through P2X7R and enhances bone resorption [29].

**Figure 3 ijms-23-04373-f003:**
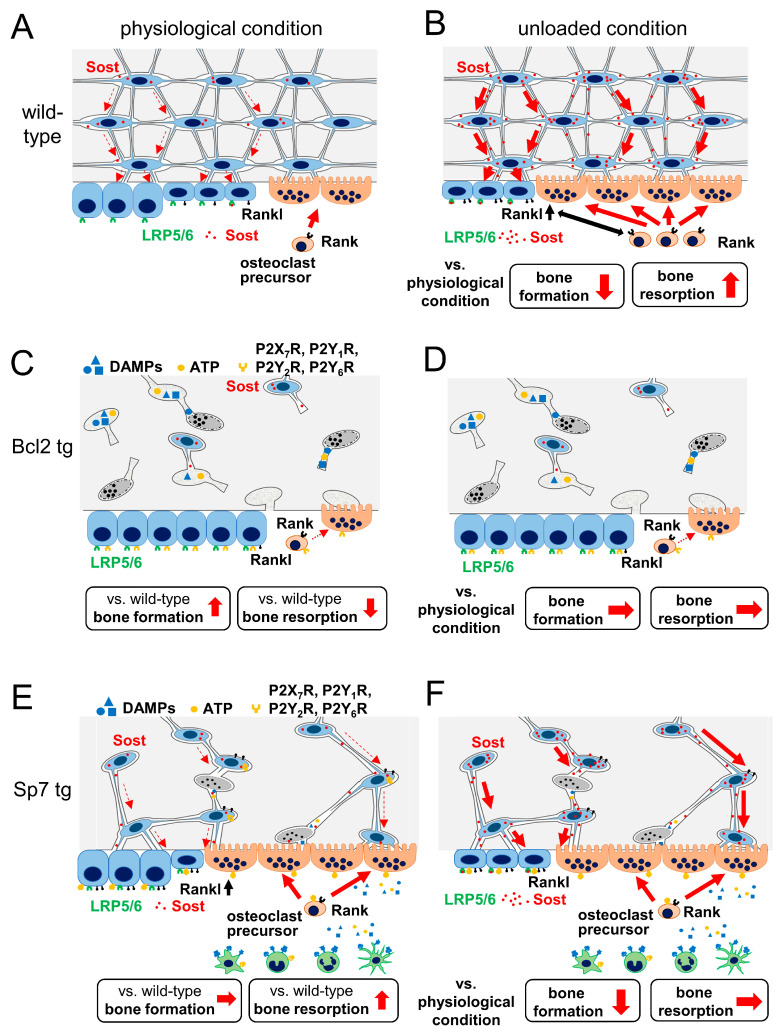
Bcl2 tg and Sp7 tg mice under physiological and unloaded conditions: In Bcl2 tg mice, the osteocyte network is disrupted by osteocyte apoptosis and their secondary necrosis due to reduced osteocyte processes and canaliculi (**A**,**C**). Sost is expressed in live osteocytes but cannot be released to the bone surface due to the interrupted lacunocanalicular network (**C**). Bone formation is increased, but bone resorption is decreased, and trabecular and cortical bone is increased in Bcl2 tg mice under physiological conditions (**A**,**C**). Sp7 tg mice have reduced osteocyte processes and canaliculi and 14% of osteocytes are TUNEL-positive (**A**,**E**). The lacunocanalicular network is functionally connected and secreted Sost reaches the bone surface (**E**). DAMPs from necrotic osteocytes, and ATP from apoptotic osteocytes are also likely to reach the bone surface and enhance bone resorption (**E**). Although bone formation in Sp7 tg mice is similar to that in wild-type mice, bone resorption is increased compared with wild-type mice due to osteocyte death under physiological conditions (**A**,**E**). In unloading, bone formation is reduced and bone resorption is increased, at least partly, through the increase in Sost and Rankl expression in osteocytes and osteoblasts, respectively, and Sost expression is increased in the compression side but not in the tension side of cortical bone in wild-type mice (**A**,**B**). Bone formation and resorption are unchanged by unloading in Bcl2 tg mice, and Sost expression in osteocytes in the compression and tension sides, and Rankl expression in osteoblasts are not altered by unloading (**C**,**D**). Sost expression is upregulated, bone formation is reduced, but bone resorption is unchanged by unloading in Sp7 tg mice (**E**,**F**). As a result, trabecular and cortical bone is reduced by unloading due to reduced bone formation in Sp7 tg mice (**E**,**F**). The dashed and solid arrows in the bone indicate a low and high amount of Sost secretion, respectively (**A**,**B**,**E**,**F**).

**Figure 4 ijms-23-04373-f004:**
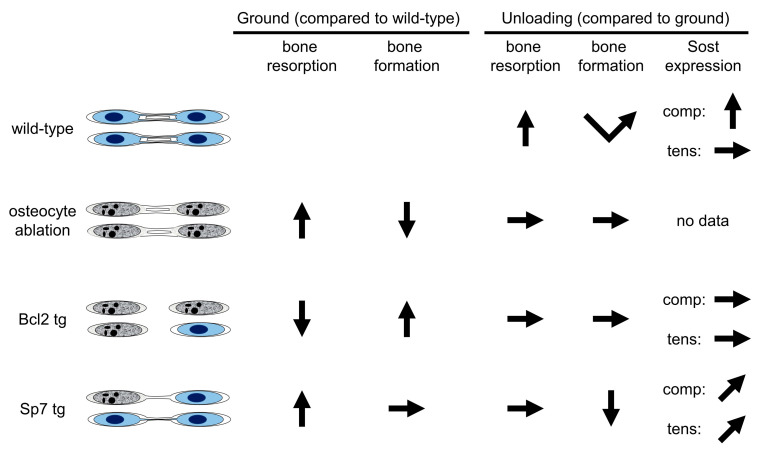
Summary of the three mouse models. The left columns show the changes in the grounded (physiological) condition compared to wild-type mice, and the right columns show the changes in the unloaded condition compared to the grounded condition. Unloading (tail suspension) experiments were performed for two weeks in wild-type, Bcl2 tg and Sp7 tg mice, and for one week in osteocyte-ablated mice [13,20,35]. In unloading of wild-type mice, bone resorption is increased and bone formation is reduced in the first week but it recovers in the second week. In osteocyte-ablated mice, bone resorption is increased and bone formation is reduced in the grounded condition due to osteocyte death, and the changes are not further altered in the unloaded condition. In Bcl2 tg mice, bone resorption is reduced and bone formation is increased in the grounded condition due to the completely interrupted lacunocanalicular network, and the changes are not further altered in the unloaded condition. Although bone resorption is increased due to osteocyte death, bone formation is normal in Sp7 tg mice in the grounded condition. Although the increased bone resorption is not further altered by unloading in Sp7 tg mice, bone formation is reduced and it never recovers during unloading. In immunohistochemical analysis, Sost expression in osteocytes is increased in wild-type mice by unloading in the compression (comp) side but not in the tension (tens) side, whereas it is unchanged and marginally increased in Bcl2 tg and Sp7 tg mice, respectively, by unloading in both sides.

## Data Availability

No available data.

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
