# Peer review of "Osteocytes: Their Lacunocanalicular Structure and Mechanoresponses"

_ijms, 2022, doi:10.3390/ijms23084373_

Round 1

Reviewer 1 Report

This review discusses the interaction of mechanical stress and lacunocanalicular structure of osteocytes. Osteocytes connect with neighbouring osteocytes through their cellular processes and that way form an extensive osteocyte network. Interesting, the lacunocanalicular network is differentially developed in the compression and tension sides of the femoral cortical bone. Three examples of transgenic mice models with an impaired lacunocanalicular structure are discussed. The authors explain the phenotypes of these mice in physiological and unloaded conditions and deduce the relationship between lacunocanalicular structure and mechanoresponses.

The review is carefully written and provides a comprehensive insight into the interaction of the osteocyte lacunocanalicular network and mechanoresponses of the cells. The manuscript contributes significant to the research field of osteocyte biology.

Major points

The rationale to include chapters about aging and lactation is not clear. Why were these topics chosen and not others, i.e. metabolic diseases as osteoporosis etc. I suggest to remove these chapters as they appear out of scope.

Figure 2: The meaning and connection of this figure to the text is difficult to grasp. What is the general message/statement of this figure? Maybe a confocal image would reflect the statement better. Maybe figure 2 and 3 could be rearranged or merged to one figure in order to better understand the meaning of the images.

The authors have chosen three different mouse models for discussing the relationship between lacunocanalicular structure and mechanoresponse. Are there other models published and if so which ones? In this line, why have the authors chosen to discuss the DTRtg, the Bcl2tg and the Sp7tg mice strains?

Minor points

Title: As the review is about osteocytes this should be reflected in the title

The introductory chapter is numbered, however, there are no follow up numbers for the subsequent chapters

Chapter “ Compression and tensions side- and gender-based differences….”: The gender part in this chapter is underrepresented and only mentioned in a sub-clause. For that it should be removed from the subtitle or gender differences should be made more clear.

Chapter “Osteocyte apoptosis and bone resorption”; again subtitle and text do not correlate. Should be better “osteocyte death” as both necrosis and apoptosis are discussed.

Figures should be mentioned in the text in the order of numbering.

Reviewer 2 Report

This review article is a unique one that focuses on a highly specific structure constructed by osteocytes in bone tissues, in which the phenotype of bone tissues of three distinct animal models is comparatively summarized and discussed.  Addressing several points would further improve its quality and readers' understandings.

Specific points

1) Since IJMS is read by a number of scientists from different areas, the title should be presented in a manner that is understandable without the background of bone research.  The present title does not tell the readers that this is a story of osteocytes.  Either "Osteocytes: Their lacunocanalicular structure and mechanoresponses", or "Lacunocanalicular structure and mechanoresponse of osteocytes" may be better. 

2) In "Introduction" section, "Lactation" emerges all of a sudden.  Please explain the role and/or biological significance of osteocytes and canalicular remodeling in lactation, and how and why enlargement of lacunae enhances mechano-responsiveness.

3) Although panel A is important, most of other panels in Figure 2 are not necessary and may rather confuse the readers.  It is recommended to combine Figures 2 and 3 into a single figure, removing a few panels in present Figure 2.  

4) In Figures 4, 5 and 6, multiple colors are used for DAMPS, which is rather confusing.  Since different DAMPS are represented by different shapes, multiple colors are redundant.  The same applies to PRRs as well.  In general, using too many colors diminish their impact.   

5) In Figure 4, osteoclast precursor should be relocated to the right to the boundary of  three osteoblasts and three osteoblasts, so that the crossing arrows could be avoided. 

6) For Figures 5 and 6, identical images are used for wild-type mice.  Combining these figures into one portrait style enables comparative recognition of these 2 mouse models, which may help better understandings of readers.  Alternatively, since the conclusion is a bit complicated, another illustration that integrates the findings obtained from the 3 models may be added as a new figure. 
